# Prognostic Significance of the Comprehensive Biomarker Analysis in Colorectal Cancer

**DOI:** 10.3390/life15071100

**Published:** 2025-07-14

**Authors:** Vera Potievskaya, Elizaveta Tyukanova, Marina Sekacheva, Zaki Fashafsha, Anastasia Fatyanova, Mikhail Potievskiy, Elena Kononova, Anna Kholstinina, Ekatherina Polishchuk, Peter Shegai, Andrey Kaprin

**Affiliations:** 1Russian Medical Academy of Postgraduate Education of Federal Scientific and Clinical Center for Specialized Medical Assistance and Medical Technologies of the Federal Medical Biological Agency, Volokolamskoye Shosse, 91, 125371 Moscow, Russia; vera.pot@mail.ru; 2P. Hertsen Moscow Oncology Research Institute—Branch or the Federal State Budgetary Institution “National Medical Research Radiological Centre” of the Ministry of Health of the Russian Federation, 2nd Botkinsky Proezd, 3, 125284 Moscow, Russia; elena24936@yandex.ru; 3 Oncology Department of Antitumor Drug Therapy, Sechenov University Clinical Hospital No.4, I.M. Sechenov First Moscow State Medical University, Str. Dovatora, 15, Bldg. 2, 119048 Moscow, Russia; fatyanova_e_s@staff.sechenov.ru (A.F.); kholstinina.anna@mail.ru (A.K.); sagekthadley@gmail.com (E.P.); 4World-Class Reserch Centre «Digital Biodesign and Personalized Healthcare», I.M. Sechenov First Moscow State Medical University (Sechenov University), Str. Bolshaya Pirogovskaya, 6, Bldg. 1, 119991 Moscow, Russia; sekacheva_m_i@staff.sechenov.ru; 5The Institute for Personalized Cardiology, I.M. Sechenov First Moscow State Medical University (Sechenov University), Str. Bolshaya Pirogovskaya, 6, Bldg. 1, 119435 Moscow, Russia; 6The Department of Oncology, Radiotherapy and Reconstructive Surgery, I.M. Sechenov First Moscow State Medical University (Sechenov University), Trubetskaya Str., Bldg. 8/2, 119991 Moscow, Russia; 7Federal State Budgetary Institution “National Medical Research Radiological Centre” of the Ministry of Health of the Russian Federation, 4, Koroleva Street, 249036 Obninsk, Russia; potievskiymikhail@gmail.com (M.P.); dr.shegai@mail.ru (P.S.); kaprin@mail.ru (A.K.); 8Peoples’ Friendship University of Russia—RUDN University, Miklukho-Maklaya Str. Bld. 6, 117198 Moscow, Russia

**Keywords:** colorectal cancer, circulating tumor DNA, violate organic compounds, metabolomics, genomics, liquid biopsy

## Abstract

Colorectal carcinoma remains one of the primary contributors to cancer deaths; however, it is also considered a preventable type of cancer, because the prognosis of the disease is directly dependent on its timely detection. Developing accurate risk prediction models for colorectal cancer is crucial for identifying individuals at both low and high risk, as risk stratification determines the need for additional interventions, which carry their own risks. The development of new non-invasive diagnostic methods based on biomaterial analysis, alongside standard diagnostic techniques such as colonoscopy with biopsy, computed tomography scanning, and magnetic resonance imaging, can address multiple objectives: improving screening accuracy, providing a comprehensive assessment of minimal residual disease, identifying patients at a high risk of colorectal cancer, and evaluating the effectiveness of ongoing treatments. The lack of sensitive diagnostic methods drives contemporary research toward the discovery of new tools for detecting tumor cells, particularly through the examination of biological materials, including blood, exhaled air, and tumor tissue itself. In this article, we analyze current studies regarding biomarkers in colorectal cancer and prognostic significance.

## 1. Introduction

Colorectal cancer (CRC) ranks as the third most common malignancy globally, representing about 10% of all cancer diagnoses and being the second leading cause of cancer-related deaths worldwide. As of February 2022, according to the data from the GLOBOCAN database, CRC accounts for over 1.9 million new cases of cancer and around 1 million deaths [1]. The burden of CRC is projected to increase by 2050 [2]. Timely identification of cancer is crucial for effective intervention, and developing diagnostic methods for asymptomatic individuals is a critical goal to improve survival rates.

Various instrumental and laboratory methods can be used for cancer diagnostics and staging, including X-ray imaging, blood tests, colonoscopy, and computed tomography (CT). However, most of these methods provide limited information regarding the presence, size, and location of pathological changes, and contradictory data may be obtained. (Table 1). Consequently, biopsy of affected tissues in most cases remains the only specific method for verifying the oncological process, although it can be challenging, complex, and expensive and carry risks of potential complications, including bleeding and even death.

Thus, biological materials, such as blood, exhaled air, saliva, and so on, nowadays are considered promising diagnostic targets for CRC detection. Biomarkers can include various types of circulating biomolecules, such as proteins, circulating tumor DNA (ctDNA), and tumor-derived cells [23]. The general strategy for identifying potential biomarkers is to create a unique, broad molecular profile/panel by analyzing specific biological molecules, including ctDNA and volatile organic compounds (VOCs), and to determine the metabolic profiles of cancer cells, as well as studying genetic alterations within tumor tissues (Figure 1). Achieving this goal may involve comparing molecular profiles in cancer patients and healthy controls, and assessing changes during treatment. In clinical oncology, this data can provide information about treatment responses, risk stratification for cancer development, and treatment methods tailored to molecular subtypes.

The use of prognostic indicators may alter threshold values for further investigation of recurrent CRC, enabling earlier interventions. Furthermore, these findings can help redefine thresholds for recommending more aggressive treatments, while exploring the genetic characteristics of tumors may provide greater prognostic sensitivity than any of the mentioned methods alone [24,25]. Thus, the key to personalized medicine in colorectal cancer lies in a comprehensive understanding of genomic and molecular data, facilitating the development of a biomarker panel with sufficient sensitivity and specificity for clinical decision-making.

## 2. Circulating Tumor DNA

Initial observations of extracellular DNA (ecDNA) date back to the 1940s, when cell-free nucleic acids were identified in the blood of cancer patients. Nonetheless, the application of ecDNA as a diagnostic biomarker in clinical practice has only been developed and implemented in the 21st century [26]. Such fragments predominantly derive from apoptotic or necrotic tumor cells, although they may also originate from viable circulating cells [27,28]. Extracellular DNA circulating in the bloodstream of cancer patients contains tumor-specific DNA sequences detectable through liquid biopsy, a novel non-invasive diagnostic method based on blood analysis (Figure 2). This represents a promising “source” of biomarkers for minimally invasive monitoring of treatment efficacy, identifying potential targets for targeted therapy, determining drug resistance, and detecting cancer at early stages.

According to the recommendations of the ESMO Precision Medicine Working Group on the clinical use of liquid biopsy, it is emphasized that plasma ctDNA analysis has advantages due to its binding to proteins, which helps protect fragments from degradation; relatively short half-life; and higher sample stability [29]. Cell-free circulating DNA harbors tumor-specific genomic modifications, including somatic mutations, aberrant methylation patterns, loss of heterozygosity, and microsatellite instability, which can serve as valuable biomarkers for cancer detection and characterization [30]. Colorectal cancer (CRC) is among the solid tumors that release the highest amounts of ctDNA into the bloodstream [31]. Thus, patients with CRC show higher levels of mutated DNA in their blood [32]. A primary limitation of this method for widespread clinical use is the lack of an optimal threshold for detecting microinvasive cancers or precancerous lesions that constitute suitable screening targets [33]. Given that the ratio of ctDNA to ecDNA can vary significantly—from less than 1% to over 40% due to various factors, including information about the primary tumor site and metastatic status—the detection of circulating tumor DNA (ctDNA) necessitates highly sensitive and specific analytical methods [34].

It is known that ecDNA is released into the bloodstream during both physiological processes and pathological non-tumor processes [35]. In patients with malignancies, cfDNA primarily derives from the tumor itself and neighboring cells, playing a role in modulating immune activity, inflammation linked to the tumor environment, and the stability of tumor growth [36]. In healthy persons, a continuous equilibrium exists between the generation and elimination of cell-free DNA (cfDNA), with the liver and kidneys playing key roles in its clearance [37].

CRC tumors have a high level of cfDNA release [38]. Despite variability in detection rates (0.01% to the majority of cfDNA) across tumor types, ctDNA levels are generally higher than circulating tumor cells, reflecting significant heterogeneity [39,40]. The tumors with low metabolic activity and a small size may be more challenging to detect, which may lead to false-negative results [41]. It is important to highlight that following radical surgery, ctDNA levels decline swiftly [42]. Despite the fact that ctDNA is a promising biomarker, and there is great potential in its usage to create a non-invasive diagnostic method, there are unresolved technical and practical problems. The two factors that hinder the accuracy of ctDNA analysis in the blood are the high rate of fragmentation and the low concentration of ctDNA during the initial stages of tumor progression [43]. ctDNA is often fragmented, making precise sequencing difficult. Chemical damage to ctDNA can further reduce the detection accuracy. Contamination of genomic DNA obtained from white blood cells further reduces accuracy, making the results less reliable [44]. Existing strategies for studying the genetic structure of a tumor require prior information about the genomic structure of the primary tumor. Also, a high concentration of ctDNA will be required for the reliable reconstruction of genome-wide changes specific to the tumor, and that reconstruction may still have low sensitivity (around 5–10% only) [45]. Digital droplet PCR can only detect one mutation per reaction [46]. NGS offers high-throughput data for a comprehensive genomic analysis, enhancing the sensitivity and accuracy in detecting genetic variations, but requires skilled bioinformaticians to detect insertions and deletions (indels), copy number aberrations (CNAs), epigenetic changes, and single-nucleotide polymorphisms (SNPs) and to assemble new genomes [47].

Currently, there are no methods of diagnosing DNA in the blood flow that have a high sensitivity and specificity and fully reflect the heterogeneity of the tumor. The lack of standardized testing protocols and the high costs associated with the current protocols are prerequisites for further research.

A potential strategy to overcome this limitation involves combining diagnostic methods, such as the detection of known cancer protein biomarkers, and using mutation panels. This approach was demonstrated in the CancerSEEK study, which included analyses of mutations from 16 genes at 1933 genomic positions and the detection of eight plasma protein biomarkers, achieving a good efficacy in identifying eight common cancer types [48]. To reduce false positives, methods target cancer-specific genomic features, such as the FDA-approved Epi proColon test, which detects SEPT9 methylation, with sensitivities of 68–72% and specificities of 80–82% [49,50]. The ColoSure test shows the methylation status of vimentin (VIM) with a comparable diagnostic accuracy [51]. Another approach to detecting cancer-specific circulating DNA is based on fragment length analysis via Fragmentomics, a validated approach for detecting seven common cancers including CRC, as employed in the DELPHI study [52]. Data obtained from studies on the methylation and fragmentation of ctDNA were incorporated into the multimodal ctDNA test LUNAR-2. The method showed an excellent diagnostic performance for stages I–II, with its sensitivity and specificity reaching 88% and 94%, respectively [53].

Surgery is the primary treatment method for patients with localized and locally advanced CRC, as well as for oligometastatic disease. Post-surgery ctDNA in the blood may signal minimal residual disease (MRD) not visible on high-resolution imaging. Evidence that ctDNA can predict recurrence in CRC patients after surgical resection for localized (Stage I-III) or oligometastatic disease has been confirmed in several studies [54,55,56,57,58,59,60].

Circulating tumor DNA plays a vital role in prognostication for patients with high-risk Stage II (T4) and Stage III colorectal cancer, providing important insights into disease progression and outcomes. Without specific prognostic markers, standard treatment includes fluoropyrimidine and oxaliplatin, though surgery alone cures about 50%, with chemotherapy offering modest survival benefits of 3–5% and 10–15% [56,57,58,59].

So, this raises the following question: Is it possible, based on ctDNA detection, to safely select only those patients who still have MRD features after surgery and who should therefore receive adjuvant treatment, while conserving economic resources by not treating patients who would not benefit from postoperative therapy? In Stage II colorectal cancer patients, the presence of ctDNA is strongly associated with an increased probability of recurrence (OR 18; 95% CI 7.9–40). Positive ctDNA correlates with worse recurrence-free survival and predicts recurrence in Stage III patients after surgery and adjuvant therapy [61]. In the IDEA study, Stage III patients with positive ctDNA had a lower 2-year RFS (64%) versus negative ctDNA (84%), indicating a worse prognosis [59].

The GALAXY study presented at ASCO 2022 validated ctDNA’s prognostic significance in over 1500 CRC patients across all stages post-surgery. Kotaka et al. found that RFS varied greatly depending on ctDNA changes from positive to negative versus positive to positive (OR 15.8) [56].

Diehl et al. [62] first reported the prognostic potential of ctDNA detection after using this method in 20 patients with oligometastatic liver involvement who underwent liver resection. Of these, 16 had detectable ctDNA post-surgery, and 15 experienced recurrences, whereas patients with negative ctDNA did not [54]. Tie et al. showed a lower RFS in liver-only metastases with positive ctDNA. Postoperative therapy cleared ctDNA in some cases, indicating treatment efficacy. In metastatic CRC, ctDNA monitoring tracks tumor molecular evolution and resistance mutations over time [58]. Two retrospective analyses demonstrated that patients with metastatic colorectal cancer (mCRC) exhibiting RAS mutations detected in ctDNA experienced markedly reduced response rates and shorter progression-free survival upon rechallenge with anti-EGFR therapies, compared to those without such mutations [58,59].

Niall J. O’Sullivan et al. [63] and Silvia Negro [64] conducted a systematic review and meta-analysis assessing ctDNA as a prognostic biomarker in colorectal cancer. The study included 22 studies involving 1676 patients. The meta-analysis showed that ctDNA positivity after neoadjuvant treatment and surgery significantly increased the recurrence risk, with hazard ratios of 8.87 and 15.15 and a pooled risk ratio of 3.66 postoperatively (*p* = 0.002). All of the above studies demonstrate the high diagnostic significance of this biomarker; however, there are a number of problems associated with the creation of a diagnostic tool with high sensitivity and specificity, taking into account the heterogeneity and biology of the tumor, as well as excluding external factors, which requires further research.

Thus, ctDNA detection represents a promising direction for numerous clinical applications, including cancer genotyping immediately at diagnosis, identifying minimal residual disease (MRD), monitoring disease progression during antitumor therapy, early detection and profiling of therapy resistance, as well as using ctDNA detection as a screening method. However, the lack of diagnostic methods with high specificity and sensitivity, and the lack of standardized thresholds and validation in various patient groups, mean we cannot consider this ctDNA a universal tumor marker; before that point, it requires further study and large-scale validation studies.

## 3. Volatile Organic Compounds (VOCs)

Recently, methods for the detection and analysis of volatile organic compounds (VOCs) have been developed as a means of increasing non-invasive diagnostic biomarkers [65]. VOCs are characterized by a low molecular weight and elevated vapor pressure. Tumor-associated VOCs can be detected in the blood, urine, and feces, on the skin, or in sweat gland secretions, as well as in exhaled breath [66,67]. One of the promising non-invasive methods for studying CRC is the qualitative and quantitative analysis of the exhaled breath for VOCs [65]. Under normal physiological conditions, the concentration of exhaled VOCs, produced because of metabolic processes, is approximately 0.9–1.2 mol/L [68].

Pathological conditions lead to specific metabolic changes that, in turn, influence the VOC profile [69]. As a new non-invasive diagnostic method for various pathological states, this approach holds great promise for cancer detection [66,70,71]. VOCs both originate from external sources, like food, drugs, and microbiome activity, and can be detected in intestinal contents, feces, blood, and exhaled air after systemic absorption and excretion [72]. Alterations in the gut microbiome in CRC influence fermentation products and organic anions, impacting both the types and amounts of VOCs produced [73], correlating with several bacterial strains that appear to exert a pro-carcinogenic effect [74]. Currently, over 800 different VOCs released from the mouth are registered in the Chemical Abstracts Service (CAS) [75]. The composition of VOCs in exhaled breath can be changed due to pathological processes, including oncological diseases. Tumor-associated inflammation, leading to increased oxidative stress and alterations in glucose metabolism and redox regulation in cancer cells, may result in distinct VOC signatures in cancer patients [76,77,78] (Figure 3).

Two primary methods are used for VOC analysis: gas chromatography–mass spectrometry (GC-MS) and electronic analyzers. The former provides a chemical analysis of specific compounds, while the latter requires “training” to recognize specific breath patterns using machine learning [79,80]. Additionally, sensor recognition systems are emerging, which detect common binding patterns of VOCs rather than individual VOCs. These systems are commonly referred to as electronic noses or E-noses [67,81].

Some potential VOCs associated with CRC have already been identified in small pilot studies [81,82,83]. Altomare et al. demonstrated that a panel of 15 VOCs detected via GC-MS effectively differentiated colorectal cancer patients from healthy individuals, achieving an accuracy rate above 80% [81].

The same research team observed that employing sophisticated breath sampling technologies—focused on isolating the alveolar fraction and eliminating environmental contaminants—enhanced the detection accuracy of 14 exhaled VOCs, achieving sensitivities of 90% and specificities of 93% [84]. Altomare DF, Di Lena M, and colleagues demonstrated that the composition of exhaled VOCs in CRC patients undergoes significant alterations following surgical intervention, reflecting metabolic changes associated with the treatment process [85]. A total of 31 VOCs were identified, which was a very good result in distinguishing pre- and post-surgery CRC patients, as well as operated patients and healthy controls. Notably, the VOC profile in recurrence-free patients after surgery exhibited distinct differences from that of healthy controls, likely reflecting altered metabolic pathways driven by oxidative stress resulting from surgical interventions and chemotherapy [85]. Conversely, Markar SR et al. observed that propanal concentrations normalized following surgery but rose again in cases of CRC recurrence, suggesting its potential role as a biomarker for disease relapse [86]. At a threshold of 28 billion−1, the sensitivity and specificity for identifying recurrence were 71.4% and 90.9%, respectively, using this volume [85,86].

Qiaoling Wang et al. evaluated the diagnostic accuracy of VOCs as a non-invasive diagnostic method for CRC in clinical practice. This comprehensive review included 32 studies, comprising 22 on VOC analysis and 9 on e-nose technology, with a total of 4688 participants. The combined sensitivity and specificity for VOC-based detection of CRC were 0.88 (95% CI, 0.83–0.92) and 0.85 (95% CI, 0.78–0.90), respectively. For e-nose methods, the pooled sensitivity was 0.87 (95% CI, 0.83–0.90) and the pooled specificity was 0.78 (95% CI, 0.62–0.88). The areas under the ROC curves for VOC analysis and e-noses were 0.93 (95% CI, 0.90–0.95) and 0.90 (95% CI, 0.87–0.92), respectively, indicating a high diagnostic accuracy for both modalities in CRC detection. The work included 10 case–control studies on VOCs in exhaled air, but all these studies had a small sample size [87].

Considering all the mentioned research results, VOCs can be considered an effective biomarker for identifying CRC patients and determining early disease recurrence, which underscores the growing interest in this field. Undoubtedly, in the coming years, breath analysis is poised to serve as a valuable adjunct in the diagnosis and monitoring of various diseases.

Although this biomarker shows significant potential, several challenges hinder its clinical application. Many VOCs originate from external sources and are influenced by environmental air levels, exposure duration, solubility, partition coefficients, as well as individual factors such as body mass and adipose tissue content [88]. Non-volatile substances, including isoprostanes and peroxynitrite, which exist as aerosols in exhaled breath, can be detected exclusively through an analysis of breath condensate [89]. Factors such as age, gender, diet, and coexisting health conditions significantly influence the composition of exhaled air, complicating the identification of low-abundance VOCs amidst numerous other compounds. Additionally, an average healthy individual exhales approximately 500 mL of breath, with about 150 mL representing dead space air originating from the upper respiratory tract [90]. Furthermore, there are no standards for breath collection techniques. Advancing breath research will require focused efforts on identifying novel biomarkers or biomarker panels, developing standardized protocols for correlating blood and exhaled breath concentrations, and establishing uniform methods for breath sample collection and storage. Additionally, distinguishing between exogenous and endogenous gases in exhaled air is essential to improve diagnostic accuracy and clinical applicability [91]. In addition to the direct identification of specific VOCs in CRC, it is crucial to establish the relationship between VOCs and the underlying disease. It remains unclear whether the metabolic processes leading to different VOC patterns are a consequence of the disease or a contributing factor to its development. Moreover, to date, there have been no prospective investigations evaluating the potential of VOCs from pre-diagnostic samples to predict the future development of colorectal cancer.

Future large, multi-center external validation trials with more robust methodologies and larger patient cohorts are needed to confirm the current data. To explore the full potential of this new method, targeted research is required to investigate the relationship between bacterial metabolism, oncological disease, and specific VOC compositions, as well as the mechanisms leading to their formation.

## 4. Metabolomics

Cancer significantly alters metabolic processes, inducing reprogramming of intracellular pathways that enable tumor cells to proliferate rapidly, adapt to the tumor microenvironment, and disrupt normal tissue metabolism [92]. Metabolomics involves the comprehensive analysis of small molecules, or metabolites (less than 1.5 kDa), within biological systems such as cells, tissues, organs, body fluids, and entire organisms, providing a snapshot of the metabolic status at a given moment [93].

This discipline encompasses the comprehensive quantification of diverse metabolites—such as nutrients, pharmaceuticals, signaling molecules, and their metabolic derivatives—within biological samples like blood, urine, tissues, or other body fluids [94]. It represents a global analysis of low-molecular-weight metabolites. Similar to other scientific disciplines, it can provide important information about disease progression. Metabolomics represents a promising frontier in oncological research. Within the framework of systems biology, a range of “omics” approaches—such as genomics, transcriptomics, and proteomics—are extensively employed to investigate the molecular underpinnings of cancer [95].

Cancer cells have their own cellular metabolic characteristics to maintain conditions for uncontrolled proliferation [96,97]. Altered metabolism results in distinctive metabolic features that may potentially become targets for drugs which selectively affect metabolic enzymes [98,99]. One of the earliest and most well-documented metabolic changes in cancer cells is their increased glucose uptake. This heightened glucose consumption can be visualized using PET-CT imaging with 18F-FDG, enabling the early detection of primary tumors, assessment of tumor burden, monitoring of the treatment response, and identification of disease recurrence. The metabolic activity of cancer cells is influenced by multiple factors, including tumor hypoxia, stromal interactions, immune cell infiltration, and genetic alterations. Moreover, genetic and epigenetic modifications may confer a survival advantage to cancer cells in nutrient-limited environments, facilitating tumor progression and resistance.

Much interest in cancer cell metabolism was initially sparked by the recognition that dysregulation of signaling pathways and reprogramming of transcription lead to metabolic changes [100]. Subsequently, research demonstrated that the activation of oncogenes and/or the inactivation of tumor suppressor genes alone can drive significant metabolic reprogramming in cancer cells [101,102]. Such metabolic changes may become a novel hallmark of cancer [96,103]. For instance, glucose, beyond being the primary nutrient for the majority of cells, also functions as an energy substrate that can initiate signaling pathways within tumor cells, thereby supporting their growth and survival [97,104,105,106]. The increased uptake of other nutrients by cancer cells promotes their survival, growth, and invasion, while their metabolic products may serve as a biomarker for cancer detection.

While glucose and glutamine are among the most frequently utilized nutrients in plasma and cell culture media, the spectrum of nutrients consumed by cancer cells and tumors is considerably broader. The specific nutrient requirements are likely influenced by the tumor subtype and the surrounding microenvironment, reflecting the metabolic adaptability of cancerous tissues. For instance, the dependence on glutamine may be influenced by high extracellular cysteine levels [107], while some tumors are less dependent on glutamine metabolism in vivo [108,109].

The tissue of origin also plays a crucial role in determining the phenotypic characteristics of cancer cells, including their metabolic profile, proliferation rate, metastatic capacity, and therapeutic response. During organ and tissue development, genetic programming guides precursor cells to differentiate into specialized cell types with distinct metabolic functions tailored to support the specific physiological roles of each tissue.

Throughout carcinogenesis, metabolic reprogramming is vital for sustaining tumor growth; however, the manner in which cancer cells and their surrounding stromal components modify their metabolism is highly dependent on the tissue type and cellular origin of the tumor [110,111]. Cancer cells may exhibit specific metabolic characteristics, akin to how they maintain markers of origin reflecting the tissue/cell of origin [112]. Furthermore, growing evidence indicates that metabolic reprogramming is also observed in various cell types within the tumor microenvironment, contributing to tumor progression and therapeutic resistance [113,114].

Metabolomics is an influential instrument capable of identifying cancer biomarkers of oncogenesis [115]. Jimenez et al. performed a comprehensive metabolic analysis comparing tumor tissues and adjacent non-tumorous mucosa obtained from patients with colorectal cancer [116]. Orthogonal partial least squares discriminant analysis (OPLS-DA) of the metabolic data revealed distinct biochemical variations between the tumor tissue samples and the adjacent mucosal tissues, highlighting metabolic alterations associated with colorectal cancer [117]. The study revealed elevated levels of choline, lactate, phenylalanine, tyrosine, taurine, and isoglutamine in tumor tissues, whereas lipid and triglyceride concentrations were reduced. These findings indicate that metabolic profiling has potential utility for detecting tumors across different stages. Additionally, Holst et al. analyzed and compared glycosylation patterns in colorectal tumor tissues versus those in non-cancerous control samples [118]. The findings revealed altered glycosylation profiles in colorectal tumor tissues, with specific modifications such as acetylation, fucosylation, sialylation, and glycan sulfation potentially serving as biomarkers for diagnosis and targeted therapy. Additionally, Tan et al. conducted serum metabolomic analyses comparing patients with colorectal carcinoma to healthy controls, highlighting metabolic differences that may aid in early detection and disease monitoring [119]. The study identified distinct metabolic alterations in patients with colorectal cancer compared to healthy individuals, including disruptions in the urea cycle, Krebs cycle, fatty acid oxidation, glutamine metabolism, and gut microbiota-related pathways. Furthermore, a specific metabolite profile was observed, characterized by increased concentrations of 2-aminobutyrate, 2-hydroxybutyrate, and 2-oxobutyrate, whereas levels of indoxyl, indoxyl sulfate, and N-acetyl-5-hydroxytryptamine were markedly decreased in the colorectal cancer group.

Maoqing Wang et al. reviewed metabolomics studies of CRC, analyzing 62 studies and 635 metabolites. They found that over 90% of these studies lacked independent validation. A total of 27 metabolites out of 635, each with an AUC exceeding 0.65 and enriched in six biological pathways, were identified. Among these, six metabolites—l-phenylalanine (reported frequency = 20), linoleic acid, citric acid, inosine, glycocholic acid, and 1-lysophosphatidylcholine—were selected based on their reported frequency. The combined AUCs for these metabolites in plasma were 0.971 (95% CI, 0.949–0.994) and 0.948 (95% CI, 0.921–0.976). Ultimately, four metabolites—l-tryptophan, linoleic acid, glycocholic acid, and 1-lysophosphatidylcholine—were prioritized due to their favorable AUCs, biological relevance, reproducibility in tumor tissues, and high reported frequency. The AUCs for these in plasma were 0.942 (95% CI, 0.901–0.983) and 0.937 (95% CI, 0.906–0.968), respectively, with a particularly high diagnostic accuracy (>0.93) for early-stage (I/II) colorectal cancer. However, it is important to note that the systematic review included various study designs—such as case reports, case–control studies, nested case–control studies, and self-controlled studies—where tumor tissue analyses often used autologous non-tumorous adjacent tissue as controls or investigated human tissues or body fluids; these methodological differences may influence the interpretation of findings [120].

Accurate and reliable methods for cancer detection are crucial not only for early diagnosis but also for screening high-risk groups, guiding initial therapeutic decisions, assessing the response to treatment, and tracking disease progression. Metabolomic analysis is based on using the metabolic hallmarks of cancer. Thus, this direction has great potential for clinical application. Determination of the metabolic profile of the tumor by its phenotyping at the beginning of the treatment process can allow patients to receive rational and personalized treatment according to their underlying molecular changes.

## 5. Genomics

Increasingly, directions such as genomics are being pursued in personalized medicine. Tumor genomics is a new science that studies the genetic characteristics of a tumor. The relationship between these two directions can potentially be used to select a drug therapy or predict the tumor response to ongoing therapy. Available tumor sequencing strategies range from targeted analysis of individual genes to targeted testing for mutations of a panel of genes, and sequencing of the entire exome or the whole genome. Genomic profiling typically employs two main strategies: sequencing of tumor tissue to identify germline or somatic mutations, with the latter being the preferred and recommended approach to minimize potential misinterpretation of genetic data.

Although numerous studies have reported conflicting results regarding molecular biomarkers, the KRAS gene currently stands as the most validated and clinically applicable marker for guiding targeted therapy with epidermal growth factor receptor (EGFR) inhibitors in patients with metastatic colorectal cancer [121]. In 2017, a panel of experts from the American Society of Clinical Pathology, the College of American Pathologists, the Association for Molecular Pathology, and the American Society of Clinical Oncology established consensus guidelines aimed at standardizing the identification of key molecular biomarkers in colorectal cancer tissues to guide epidermal growth factor receptor (EGFR)-targeted therapies and conventional chemotherapy protocols [122]. The panel reviewed over 4000 scientific publications and determined that genetic alterations within the EGFR signaling pathway are predictive of resistance to EGFR-targeted treatments in colorectal cancer patients [123].

The accumulating data indicate that a significant role in the pathogenesis of cancer is played by a malfunction of transcription enhancers [124,125]. The interaction of both gene mutations and epigenetic changes can also cause the progression of malignant adenocarcinomas, which is driven by disruptions in signaling pathways that regulate cellular proliferation and tumor advancement [126,127] (Figure 4).

Recent studies have shown a cross-connection between gene mutations, such as BRAF, KRAS, p53, and MSI, epigenetic changes, and DNA methylation of the promoter regions of CpG islands in the development of cancer. Indeed, genetic mutations may influence epigenetic modifications, while epigenetic alterations can, in turn, impact mutational dynamics and contribute to genomic instability [130].

Tumor sequencing aimed at identifying somatic mutations typically involves analyzing DNA extracted from blood or saliva samples, with germline DNA serving as a reference genome to distinguish somatic alterations from inherited variants [131]. Germline DNA profiling during this process allows for the detection of diverse genetic variants, which are often considered incidental or secondary findings. Although these variants are not directly related to the primary purpose of tumor mutation analysis, they hold significant clinical relevance and provide valuable medical insights [132]. In cases of accidental detection of suspected pathogenic variants of the germinal line (PVGL), which cause a hereditary predisposition to cancer, clinical and genetic counseling should be conducted [133].

Genomic profiling in colorectal cancer is aimed at identifying somatic mutations, while random changes in the germinal line can also be detected. Recent research indicates that comprehensive exome sequencing aimed at identifying somatic mutations may result in a higher detection rate of germline mutations among patients compared to conventional targeted germline testing guided by clinical criteria and the family history [132]. The guidelines emphasize the benefits of profiling mutations in KRAS/NRAS, BRAF genes, and DNA mismatched repair (MMR) among patients with metastatic CRC [134]. However, the study of germinal mutations is most often kept out, although a family or hereditary component can be detected in almost 25% of all cases of CRC [135]. Y. Nancy You, M.D., M.H.Sc. et al. investigated germline variants in patients with advanced colorectal cancer who underwent tumor sequencing for somatic mutations. They analyzed and compared the molecular profiles, clinical characteristics, and ancestral backgrounds of patients with suspected germline mutations to better understand their implications for disease progression and hereditary risk [131]. The researchers suggested that tumor sequencing may reveal more germinal changes than traditional germline testing based on clinical criteria and pedigree. The study identified a total of 1910 germline variants across 151 patients. Following further analysis, 15 pathogenic germline mutations were confirmed in 15 individuals, involving nine genes associated with varying degrees of penetrance in colorectal cancer. These included CHEK2 (4; 27%), APC (2; 13%), BRCA1 (2; 13%), CDH1 (2; 13%), MSH2 (1; 7%), MSH6 (1; 7%), NF2 (1; 7%), TP53 (1; 7%), and ATM (1; 6%). Patients harboring pathogenic germline variants tended to be diagnosed at a younger age compared to those without such mutations (median 45 versus 52 years; *p* = 0.03). Among the 15 individuals with confirmed pathogenic mutations, nearly half (7 patients; 46.7%) carried variants in genes associated with low to moderate penetrance for colorectal cancer, and notably, 2 of these patients (28.5%) possessed clinically significant variants in CDH1 and NF2 that may have been overlooked by standard clinical criteria and pedigree-based testing [131]. While the Cancer Genome Atlas (TCGA) reports mutually exclusive mutations such as APC versus PIK3CA and KRAS versus FOLR3, observations in a Caucasian patient cohort revealed that PIK3CA and KRAS mutations were most prevalent, with PIK3CA mutations notably mutually exclusive with TP53 mutations [136]. In contrast, the Taiwanese cohort demonstrated a higher frequency of PIK3CA and KRAS mutations in tumors located in the left colon, whereas APC mutations were predominantly found in right-sided cancers. These findings suggest distinct driver gene involvement and alternative carcinogenic pathways between Caucasian and Taiwanese populations. Additionally, metabolomic profiling of blood samples using chemometric techniques uncovered significant differences in metabolic signals between patients with right-sided and left-sided colorectal cancer, highlighting molecular heterogeneity linked to tumor location [136].

In addition to the most well-known mutations in the KRAS, NRAS, and BRAF genes and the detection of microsatellite instability, currently, more attention is being paid to other promising epigenetic markers. The concept of “epigenetics” encompasses heritable modifications that alter the gene expression or phenotype without involving alterations in the underlying DNA sequence. Among these, DNA methylation has emerged as one of the most extensively investigated biomarkers in colorectal cancer, serving a crucial function in modulating gene activity during the process of carcinogenesis [137]. Hypermethylation of CpG islands within promoter regions of tumor suppressor genes is a recognized epigenetic mechanism leading to gene silencing and contributing to tumor development [138]. Gene transcription inactivation results from alterations in chromatin architecture at gene promoters, rendering them inaccessible to transcription factors. These epigenetic modifications can disrupt multiple cellular pathways, including DNA repair mechanisms, apoptosis, angiogenesis inhibition, metastasis suppression, cell cycle regulation, and cell adhesion. Notably, methylation of the VIM gene is detected in a significant proportion of colorectal cancer cases (ranging from 53% to 84%). The detection method employs PCR technology to quantify methylated IN DNA and assess DNA integrity, achieving high sensitivity (83%) and specificity (82%) in diagnostic applications [139]. A recent meta-analysis encompassing 25 studies by Nian et al. demonstrated that methylated SEPT9 (Epi ProColon; Epigenomics AG) serves as a reliable blood-based biomarker for colorectal cancer detection, particularly showing high efficacy in identifying advanced-stage tumors [140]. Perez-Carbonell et al. performed a comprehensive evaluation of a panel comprising CRC-specific methylated genes (SEPT9, TWIST1, IGFBP3, GAS7, ALX4, and miR137) and observed that methylation levels of all these markers were markedly elevated in colorectal cancer tissues compared to normal controls (*p* < 0.0001). Notably, the highest methylation frequencies were detected for miR 137 (86.7%) and IGFBP 3 (83%), underscoring their potential utility as biomarkers for CRC detection. The combined assessment of these two genes achieved a sensitivity of 95.5% and a specificity of 90.5% for colorectal cancer detection, indicating its potential as a valuable diagnostic biomarker. Additionally, the study revealed that hypomethylation of IGFBP 3 may serve as an independent prognostic factor associated with poorer outcomes in patients with stage II and III CRC. Notably, among patients with stage II and III disease exhibiting IGFBP3 hypermethylation, the administration of adjuvant 5-fluorouracil (5-FU) chemotherapy did not lead to significant improvements in overall survival or disease progression, highlighting the importance of epigenetic profiling in guiding therapeutic decisions [141].

Currently, the molecular genetic classification of CRC subtypes (Consensus Molecular Subtypes) is used, based on complex profiling of gene expression. Colorectal cancer can be classified into four distinct subtypes, designated as CMS1, CMS2, CMS3, and CMS 4, each characterized by specific biological features and gene expression profiles [122]. In a recent retrospective study conducted by Okita et al., 193 patients with metastatic colorectal cancer were studied. The results of the study showed that the biological features of the molecular genetic subtypes of the tumor can have a significant impact on the effectiveness of chemotherapy. The study findings indicated that, within the CMS4 subtype, chemotherapy protocols incorporating irinotecan yielded superior outcomes compared to oxaliplatin-based regimens, with significant improvements in progression-free survival (hazard ratio [HR] = 0.31; 95% confidence interval [CI]: 0.13–0.64) and overall survival (HR = 0.45; 95% CI: 0.19–0.99) [142]. Concerning anti-EGFR therapies, patients classified as CMS1 exhibited the poorest prognosis, with an increased risk of disease progression (HR = 2.50; 95% CI: 1.31–4.39) and mortality (HR = 4.23; 95% CI: 1.83–9.04). Conversely, CMS2 patients demonstrated the most favorable outcomes, showing the best progression-free survival (HR = 0.67; 95% CI: 0.44–1.01) and overall survival (HR = 0.49; 95% CI: 0.27–0.87) relative to other molecular subtypes [142].

The latest findings in research are contributing to a deeper comprehension of the intricate processes that underlie genetic alterations in colorectal cancer. The molecular genetic profiling of colorectal cancer holds significant importance not only for accurately predicting the progression of the disease but also for evaluating the efficacy of proposed anticancer drug therapies. In situations of uncertainty, it serves as a crucial tool for meticulously weighing the potential risks and benefits when formulating treatment plans. Innovative treatment strategies are being actively developed, particularly for advanced stages of the disease. One of the most promising advancements in CRC treatment is the ability to detect mutations or their absence in genes such as KRAS, BRAF, and NRAS, which allows for the selection of targeted therapies. However, complex clinical scenarios necessitate a deeper exploration of mechanisms of drug resistance, treatment failure, and disease recurrence.

The process of cancerogenesis is based on the continuous accumulation of mutations. Malignant cells exhibit an increased mutation rate and genetic instability. Genetic instability enhances alterations in tumor cells, contributing to mutation accumulation, and metabolomic, proteomic, and morphological changes [143]. Various cancer cells with different genetic and phenotypic features interact with each other, the tumor microenvironment, and the immune system. The process is known as tumor clonal evolution and results in an increase in tumor malignancy, cancer progression, and resistance to drug treatment and radiotherapy [144]. Thus, the evaluation of various biomarkers with the implementation of the evolutionary approach is required for the future development of personal strategies for colorectal cancer treatment.

Currently, the obtained data have enabled the development of DNA methylation-based diagnostic panels that can be applied in clinical practice. The key features and advantages of such panels include their pan-cancer potential, characterized by distinct methylation patterns that allow for broad-spectrum oncological detection.

Clinomics Inc. is developing panels for the detection of oncological diseases based on DNA methylation profiling, utilizing epigenetic markers to enhance diagnostic accuracy [145,146,147]. Fundamental data on characteristic methylation patterns across various cancer types have enabled Clinomics Inc. to identify the most informative genomic regions for the development of their diagnostic panels.

GRAIL Gallery employs methylation pattern profiling for comprehensive blood-based multi-cancer screening, capable of detecting more than 50 cancer types at early stages. The technique focuses on identifying circulating tumor DNA characterized by abnormal methylation signatures. Initial studies indicate that the test’s sensitivity for early-stage cancers (Stages I–II) is around 27.5%. These results were derived from symptomatic patient cohorts and may not fully represent the test’s performance in asymptomatic screening settings. When the analysis was restricted to 12 specific cancer types identified as high-priority for early detection, the sensitivity improved to 52.8%, highlighting the potential for targeted multi-cancer screening approaches [148]. Nevertheless, several unresolved issues persist, including the risk of false-positive findings and their potential impact on patients and the healthcare system. Additionally, there is ongoing debate about whether a single negative test result is sufficient to definitively exclude the presence of disease or if periodic retesting is necessary to ensure accurate diagnosis and monitoring.

The MethylationEPIC BeadChip is employed for high-throughput genome-wide DNA methylation profiling, covering over 850,000 CpG sites, including promoters, enhancers, and other regulatory regions. In the field of diagnostics, it facilitates the detection of epigenetic alterations that are characteristic of various diseases, including different types of cancer [149,150]. Although the capabilities for epigenetic analysis are substantial, several limitations remain, including restricted genome coverage, potential selection bias toward specific genomic regions, reduced sensitivity in samples with low methylation levels, and the high costs associated with conducting such studies [151,152].

DNA methylation-based panels represent a powerful tool for early cancer detection due to the stability of epigenetic modifications and their tumor-specific patterns. These panels are continuously being refined and integrated into clinical workflows, with ongoing efforts to expand their applicability across a broader range of oncological conditions. Such advancements enable earlier diagnosis and contribute to improved treatment outcomes. However, each method has inherent limitations, including challenges related to standardization and data interpretation. The analysis often requires sophisticated, costly equipment and complex data processing, which currently restricts widespread adoption in routine clinical settings.

## 6. Novel Biomarkers

In cancer treatment, processes above the organizational level of the genome and the transcriptome must also be targeted. Other molecular targets can be used as new approaches in the identification of biomarkers for cancer. Immune-related proteins, exosome-based markers, and microbiome-based biomarkers have become increasingly of scientific interest in recent years.

Feces are considered an indicator of gut health and can provide information about the pathological processes occurring in them. Tumor progression may compromise the integrity of the intestinal epithelium, resulting in infiltration by immune cells that can disrupt the intestinal barrier, even during the early stages of colorectal cancer [153].

Gut microbiota, including species such as *Alistipes*, thereby exacerbate tumor-associated inflammation and potentially influence disease progression [154]. Also, the gut bacteria, such as *Alistipes* species [155], can stimulate host immune cells and promote tumor-associated inflammation, which may contribute to cellular mutations and facilitate tumor progression [156]. Consequently, an increased presence of immune cells and related proteins in the stools of CRC patients may serve as potential biomarkers for non-invasive diagnosis and disease monitoring.

Hao Zhang and colleagues developed a novel fecal biomarker panel consisting of five immune-related proteins—CAT, LTF, MMP9, RBP4, and SERPINA3—that exhibit significant diagnostic accuracy for colorectal cancer (CRC). These proteins were markedly overexpressed in tumor tissues and showed a positive correlation with the levels of CRC-associated microbiota. This research emphasizes the potential application of fecal immune biomarkers as non-invasive diagnostic tools for CRC and highlights the intricate relationship between tumor immune responses and gut microbiota composition [157].

Several studies demonstrate the role of the immune-related proteins such as catalase [158], lactoferrin [159], matrix metalloproteinase-9 [160], retinol binding protein 4 [161], and serine protease inhibitor A family member 3 [162] in metabolic pathways and in the regulation of cellular processes, modulation of the host immune system, development, progression, and metastasis in CRC.

Other biomarkers, such as extracellular vesicles (EVs), can be used for cancer diagnosis. It is a novel approach for cancer diagnosis. Extracellular vesicles are a diverse group of lipid bilayer-enclosed particles that are actively produced and released by various cell populations into the extracellular space. These vesicles function as carriers for a wide range of cellular components, playing a crucial role in intercellular communication and supporting numerous biological functions. Vesicles are heterogeneous particles enclosed in a lipid bilayer. They are actively synthesized and secreted by many cells of the body and serve as conduits for transporting cellular components, such as lipids, nucleic acids, proteins, and various kinds of RNAs, including messenger RNA, microRNA, circular RNA, long non-coding RNA, and others. This is one of many ways of intercellular communication and participates in a multitude of biological processes [163]. The clinical relevance of EVs is different between types of cancers [164]. There is growing evidence that EVs are associated with colorectal cancer [165].

A meta-analysis by Xianquan Shi et al., which included 48 studies, was performed to investigate the diagnostic significance of EVs. The combined analysis of EV-derived RNAs demonstrated a sensitivity of 76%, specificity of 75%, and AUC of 0.87, while EV RNA panels achieved a sensitivity of 82%, specificity of 79%, and AUC of 0.90. Similarly, individual EV protein markers showed a sensitivity of 85%, specificity of 84%, and AUC of 0.92, with EV protein panels reaching 87% sensitivity, 83% specificity, and an AUC of 0.92. Notably, in early-stage CRC, EV-based biomarkers exhibited promising diagnostic performance, with sensitivity at 80%, specificity at 75%, and an AUC of 0.89 [166].

In another meta-analysis by Zilong Wu et al., 28 studies were analyzed. The meta-analysis revealed that across 29 studies, the pooled sensitivity was 0.74 (95% confidence interval [CI], 0.70–0.78), while the pooled specificity was 0.81 (95% CI, 0.78–0.83), indicating a moderate diagnostic accuracy of the evaluated biomarkers [167].

Also, high diagnostic efficacy was confirmed in the meta-analysis by Shu-ya Liu et al. The pooled analysis demonstrated an overall sensitivity of 0.62 (95% CI: 0.60–0.63), specificity of 0.76 (95% CI: 0.75–0.78), positive likelihood ratio (PLR) of 3.07 (95% CI: 2.52–3.75), negative likelihood ratio (NLR) of 0.34 (95% CI: 0.28–0.41), and diagnostic odds ratio (DOR) of 10.98 (95% CI: 7.53–16.00). The summary receiver operating characteristic (SROC) curve yielded an area under the curve (AUC) of 0.88, supporting the high potential of extracellular vesicles as reliable early diagnostic biomarkers for colorectal cancer [168].

Circulating biomarkers demonstrate significant promise for the diagnosis of colorectal cancer and could play a crucial role in guiding therapeutic strategies; however, additional research is necessary to establish their practical utility in clinical settings.

After analyzing the literature data, we compiled a summary table for each of the biomarkers, indicating the advantages and disadvantages of each of them (Table 2).

The prospects for the application of biomarkers based on proteomic, genomic, and microbiome analyses, utilizing ctDNA, VOCs, and exosomes in the diagnosis of colorectal cancer are significantly expanding due to advancements in multi-omics technologies and the integration of artificial intelligence methods. The combination of data from various molecular levels enables a more precise and comprehensive understanding of the biological alterations associated with the disease, thereby facilitating earlier and more reliable diagnosis. 

A crucial role in this process is played by the development of modern approaches grounded in multi-omics, which allow for the detection of molecular abnormalities at early stages with high sensitivity. The implementation of machine learning algorithms and AI systems ensures automated processing of large datasets, identification of complex patterns, and the development of personalized diagnostic models. In the future, the integration of these technologies promises to revolutionize screening and monitoring strategies for colorectal cancer, making them less invasive, more accurate, and accessible to a broader patient population.

## 7. Cost-Effectiveness of New Biomarkers

In recent years, significant focus has been placed on assessing the economic viability of different biomarker-based diagnostic approaches, such as circulating tumor DNA (ctDNA), volatile organic compounds (VOCs), metabolomics, genomics, exosomes, and immuno-oncological proteins, in the diagnosis of colorectal cancer (CRC). This analysis explores the potential of these biomarkers to reduce overall treatment costs by enabling early detection of the disease and decreasing the need for expensive therapies in advanced stages. The comparative assessment indicates that the implementation of innovative biomarker-based strategies can significantly enhance the economic efficiency of diagnostic protocols and alleviate the financial burden on healthcare systems. Numerous studies indicate that the use of ctDNA as a tool for early detection of CRC has significant potential to reduce overall treatment costs for patients with this disease. Specifically, research findings demonstrate that integrating ctDNA into screening programs enables the detection of cancer at earlier stages, which substantially decreases the need for costly therapies associated with advanced disease. 

A recent systematic review conducted an economic evaluation of ctDNA for cancer screening. It was shown that mSEPT9 proved to be more cost-effective both in comparison with traditional screening methods and in comparison with the lack of screening [180]. This is primarily because early diagnosis facilitates more effective and less expensive treatment options while also reducing the risk of complications and the necessity of repeated hospitalizations. Therefore, incorporating ctDNA into diagnostic strategies could represent an important step toward enhancing the economic efficiency of healthcare systems in the fight against colorectal cancer. 

Volatile organic compounds (VOCs) represent a promising and innovative biomarker for the diagnosis of colorectal cancer (CRC). Research in the field of medical diagnostics indicates that breath analysis for the presence of specific VOCs can serve as a non-invasive, safe, and relatively cost-effective method for early disease detection. This approach is based on the fact that tumor processes in the body can lead to alterations in metabolic pathways, resulting in the emission of certain VOCs that can be detected in exhaled air. Utilizing VOC analysis for CRC diagnosis reduces the need for more invasive and expensive procedures, such as colonoscopy or biopsy, and facilitates earlier detection of the disease [181,182]. Daniah Alsaadi et al. demonstrated combined research findings, where six of the most frequently detected VOCs in patients with colorectal cancer (ethanol, acetone, ethyl acetate, 4-methyloctane, nonanal) were identified. The combined sensitivity and specificity of these VOCs were 0.89 (95% CI = 0.80–0.99) and 0.83 (95% CI = 0.74–0.92), respectively. The potential value of this biomarker has been demonstrated. However, currently, its application is limited due to the lack of standardized analytical platforms, small patient sample sizes in studies, and the absence of multicenter randomized validation trials, which prevents its widespread use as a diagnostic tool in clinical practice [183]. Metabolomic and genomic analyses offer critical insights into the underlying metabolic processes and genetic changes linked to colorectal cancer (CRC). These techniques have the potential to enhance early detection and improve prognostic assessments. Current research indicates that incorporating metabolomics and genomics into routine clinical workflows could lead to an estimated 20% reduction in treatment expenses by enabling more targeted and personalized therapeutic interventions [184,185,186]. There has been a sustained increase in interest in the use of exosomes and immuno-mediated proteins as biomarkers for colorectal cancer (CRC) screening. Studies indicate that exosomal analysis enhances the sensitivity and specificity of early detection, which in turn can contribute to reducing healthcare costs associated with advanced-stage disease management. The implementation of combined assays incorporating exosomal markers has the potential to decrease overall diagnostic and treatment expenses for CRC by approximately 15–20%, owing to more accurate identification of precancerous lesions and early-stage malignancies. Additionally, evidence suggests that immuno-mediated proteins, such as cytokines and surface markers of immune cells, can further improve screening efficacy by detecting immunological alterations during the initial phases of tumor development [187,188,189,190,191].

An analysis of potential economic efficiency indicates that the use of these biomarkers in colorectal cancer diagnosis could significantly reduce treatment costs associated with advanced-stage malignancies. Incorporating these technologies into clinical workflows may enhance patient outcomes while contributing to a more efficient allocation of healthcare resources.

## 8. Prospects of Using Multimodal Diagnostics Using Artificial Intelligence

The prospects of artificial intelligence (AI) and machine learning (ML) in modern medicine are undeniable. In recent years, advancements in these fields have opened up new horizons in the diagnosis of oncological diseases, with colorectal cancer being no exception. Existing invasive diagnostic methods, such as colonoscopy, combined with computer-aided detection techniques have already contributed to reducing the miss rate of adenomas and the risk of cancer development by improving screening outcomes [192]. In recent years, advancements in these fields have opened up new horizons in the diagnosis of oncological diseases, with colorectal cancer being no exception. Existing invasive diagnostic methods, such as colonoscopy, combined with computer-aided detection techniques have already contributed to reducing the miss rate of adenomas and the risk of cancer development by improving screening outcomes [193]. The utilization of computer-aided detection systems (CADe) and AI will enhance both the speed and accuracy of colorectal cancer detection [194]. The integration of artificial intelligence into endoscopic diagnostics is anticipated to substantially enhance diagnostic accuracy, thereby reducing the incidence of false-positive and false-negative results. This improvement is expected to decrease unnecessary interventions and overtreatment, ultimately contributing to an improved cost-effectiveness within healthcare systems. The adoption of AI-driven tools in endoscopy has the potential to optimize clinical decision-making, improve patient outcomes, and streamline resource utilization, marking a significant advancement in the efficiency of oncological screening [195].

Non-invasive diagnostic techniques primarily rely on the detection and analysis of tumor-specific biological markers, including circulating tumor cells, ctDNA, exosomes, and other molecular entities. The key benefits of these approaches are their minimally invasive nature and the rapidity of testing, which significantly enhance patient comfort and adherence to screening protocols. Additionally, such methods contribute to reducing healthcare expenditures by enabling earlier diagnosis, continuous disease monitoring, and personalized treatment adjustments. Their integration into clinical practice holds great potential for improving oncological outcomes through more accessible and patient-centered diagnostic strategies [196]. Despite ongoing advancements, the current limitations of individual biomarkers prevent the identification of a single marker with optimal sensitivity and specificity for clinical application. To address this challenge, recent decades have seen a surge in the development of multi-omics diagnostic approaches, which aim to enhance the diagnostic accuracy of existing biomarkers and effectively filter out less informative candidates. These integrative strategies hold the promise for overcoming the inherent constraints of standalone biomarkers, paving the way toward more reliable and precise diagnostic tools in personalized medicine [197]. Machine learning models based on the analysis of comprehensive biomarker profiles demonstrate significant potential for improving the accuracy of detecting precancerous lesions and early-stage CRC. Current research indicates that algorithms such as convolutional neural networks (CNNs) and gradient boosting methods effectively integrate data on circulating ctDNA mutations, metabolic profiles, and protein expression levels to develop diagnostic models with sensitivities reaching up to 90% and specificities exceeding 85%. These approaches enable more a precise identification of early disease markers, facilitating timely intervention and enhancing the patient prognosis [198]. The application of artificial intelligence for the analysis of circulating RNA and immuno-mediated proteins has gained particular importance in modern early diagnostic approaches, enabling the identification of unique immune response patterns at initial disease stages. Recent studies have demonstrated that the integration of microRNA, circulating cell-free DNA (cfDNA), and immune protein profiles through machine learning techniques significantly enhances screening accuracy. Specifically, the combined analysis of cfDNA alongside microRNA and immunological markers allows for more the reliable detection of precancerous lesions and early-stage malignancies, surpassing the efficacy of conventional diagnostic methods [199,200,201,202].

The integration of machine learning and artificial intelligence systems into the healthcare framework for non-invasive colorectal cancer screening offers significant economic benefits. According to recent analytical reviews, the utilization of automated risk assessment models substantially reduces the number of unnecessary colonoscopies, thereby decreasing diagnostic costs and alleviating the burden on healthcare resources. Furthermore, earlier detection of the disease contributes to lowering expenses associated with the treatment of advanced-stage cancers [203,204,205]. Additionally, cost savings are achieved through the automation of test result interpretation and improved diagnostic accuracy, which reduces the incidence of false-positive cases and the need for subsequent procedures. Overall, economic modeling indicates a favorable return on investment in the development and implementation of AI systems within national screening programs.

The application of artificial intelligence-driven multi-omics analysis offers a transformative potential for advancing precision medicine through the integration of diverse biological datasets. However, its practical implementation faces several significant challenges. A primary obstacle is the heterogeneity and lack of standardization across omics data, as datasets are often generated using different platforms, protocols, and populations. This variability introduces inconsistencies that complicate the training, validation, and generalization of AI models, thereby limiting their clinical utility and reproducibility [206].

Future directions in this field include the development of multi-omics platforms leveraging big data analytics, as well as the implementation of unsupervised learning techniques to discover novel biomarkers. A critical goal is the creation of versatile algorithms capable of adapting to diverse populations and accounting for individual patient variability. Overall, the integration of multi-omics data with artificial intelligence-driven approaches for comprehensive biomarker profiling holds significant promise for enhancing the efficacy of non-invasive colorectal cancer diagnostics. Such advancements are expected to not only improve the screening accuracy but also reduce healthcare costs by enabling earlier disease detection and streamlining diagnostic workflows, ultimately contributing to more personalized and cost-effective patient care.

## 9. Conclusions

High mortality rates from CRC determine the need to find new solutions. The paradigm of cancer non-invasive diagnosis is undergoing a significant shift with the appearance of biomarkers. Advanced “omics”-based platforms have emerged as highly effective tools for identifying cancer biomarkers. Comprehensive analysis of immune-related markers combined with multi-omics approaches holds promise for precise, non-invasive diagnosis, potentially leading to significant advancements in personalized cancer therapy.

Lack of standardization in methodology and sample identification could reduce errors in diagnostic biomarkers and promote their clinical feasibility. Combining all the above-mentioned biomarkers in a multi-omics approach could enhance cancer detection and monitoring. This approach offers a more comprehensive strategy for CRC surveillance. Conducting large-scale, multicenter, external validation studies based on comprehensive biomarker profiling will reduce the impact of interfering factors due to the lack of standardized thresholds and validation in various patient groups and will allow this diagnostic method to be applied in clinical practice.

## Figures and Tables

**Figure 1 life-15-01100-f001:**
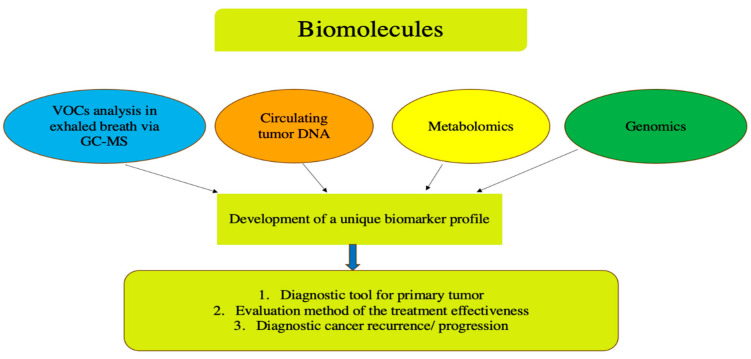
Schematic representation of the comprehensive biomarker analysis in colorectal cancer with a focus on non-invasive diagnostic methods. VOCs—violate organic compounds.

**Figure 2 life-15-01100-f002:**
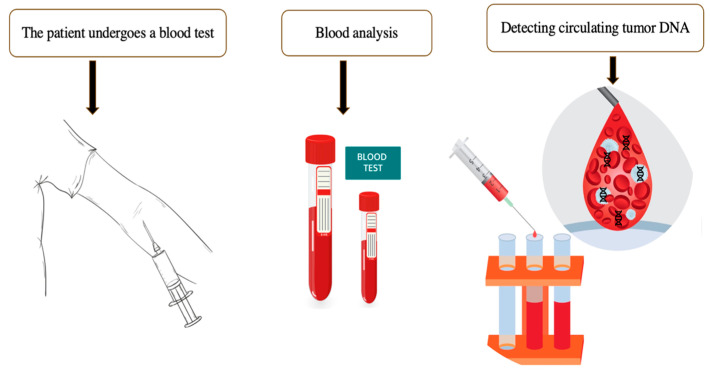
The algorithm for performing a liquid biopsy. Extracellular DNA circulating in cancer patients can be detected in the bloodstream. The presence of circulating DNA may indicate the presence of tumor cells in the human body even before clinical, endoscopic, and radiological manifestations.

**Figure 3 life-15-01100-f003:**
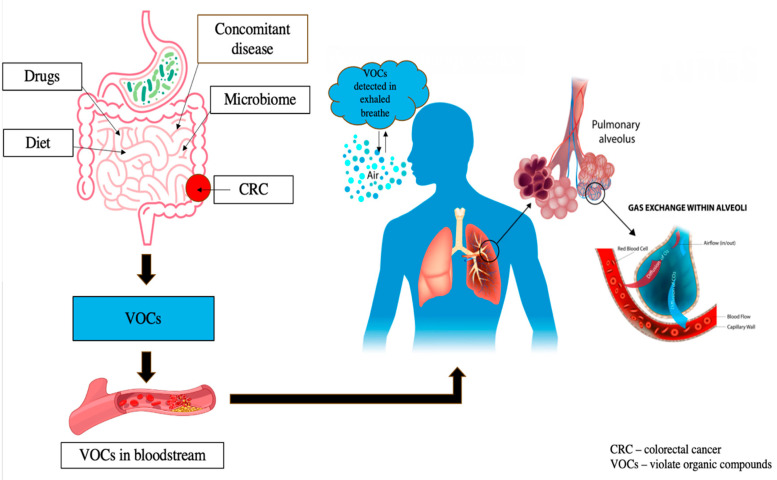
VOC profiles are influenced by external factors including food, drugs, and microbiota activity, as well as internal conditions like comorbidities. After production in the gut, these compounds circulate through the bloodstream to the lungs, allowing their measurement in exhaled air for potential diagnostic applications.

**Figure 4 life-15-01100-f004:**
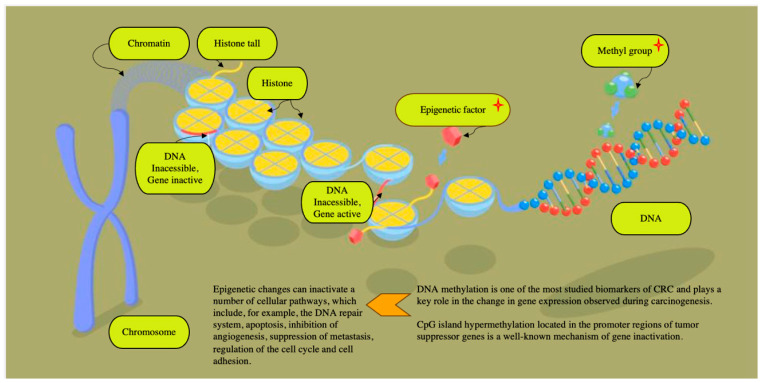
The pathogenesis of colorectal cancer involves a multitude of complex mechanisms, which include, among others, chromosomal instability (CIN), the phenotype of methylation of CpG islands (CIMP) and microsatellite instability (MSI) [128]. Examining DNA methylation have concluded that there are at least three subtypes of CRC, depending on the rate of DNA methylation and mutations in key genes [129].

**Table 1 life-15-01100-t001:** Summary table of standard diagnostic methods for CRC.

Standard Diagnostic Methods	Sensitivity	Specificity	Advantages	Disadvantages
gFOBT * (Guaiac fecal occult blood test) (Hemoccult Sensa, Beckman Coulter)	7–21% [3]	50–75% [3]	Ease and simplicity of testing. Availability as a deliverable method. Non-invasiveness. Cost-effectiveness. Allows detection of potential sources of bleeding, which facilitates early initiation of a comprehensive follow-up examination [3,4,5,6].	High frequency of false-positive and false-negative results. Does not allow for accurate identification of the source of bleeding or the nature of the disease. Poorly informed in the early stages of the tumor process. Limited diagnostic accuracy. Special requirements for the preparation and collection of samples. Patient testing preparation is required [6,7].
FIT * (Fecal immunochemical test) (OC-Sensor and OC-Light; Polymedco)	25–27% [3]	74–81% [3]	Ease and simplicity of testing. Availability as a deliverable method. Non-invasiveness. Cost-effectiveness. Allows detection of potential sources of bleeding, which facilitates early initiation of a comprehensive follow-up examination. No patient preparation is required [7,8].	High frequency of false-positive and false-negative results. Does not allow for accurate identification of the source of bleeding or the nature of the disease. Poorly informed in the early stages of the tumor process. Limited diagnostic accuracy. Special requirements for the preparation and collection of samples [7,9].
DNA-FIT * (Fecal Immunochemical test combined with DNA testing)	47% [3]	93% [3]	Simplicity of testing. Availability as a deliverable method. Non-invasiveness. Allows detection of precancerous diseases. No patient preparation is required [4,7,10,11].	The necessity of using invasive diagnostic procedures to confirm the diagnosis. Less sensitivity to polyps without malignant transformation. High frequency of false-positive results due to inflammatory bowel disease. Cost-intensive [4,7,11,12].
Colonoscopy *	95% [3]	86–89% [3]	High sensitivity and specificity. The possibility of simultaneous removal of polyps. Detection of precancerous changes. Regular colonoscopy significantly reduces the incidence and mortality of colorectal cancer [13,14].	Invasiveness and risk of complications. Dependence on the experience of an endoscopist. Special requirements for study preparation. Cost and availability [4,7,12].
Flexible sigmoidoscopy *	95% [3]	87% [3]	High sensitivity and specificity. Detection of precancerous changes. Regular sigmoidoscopy significantly reduces the incidence and mortality of colorectal cancer [4,15].	Invasiveness and risk of complications. Limited intestinal examination area. The inability to remove polyps or perform a biopsy. The requirement for repeated diagnostic testing [4,7,14].
Computed tomography (CT) colonography *	86–100% [3]	86–98% [3]	Non-invasive method. High sensitivity for large polyps and cancer. The procedure takes less time compared to a traditional colonoscopy. Possibility of evaluation of other abdominal organs. Suitable for patients with contraindications to invasive methods [4,7,11,16].	Suitable for patients with contraindications to invasive methods. The need for preliminary preparation of the intestine. The inability to remove polyps during the study. Radiation exposure [4,7].
CT scan	70–85% (depending on the stage of the disease and the technique used) [17,18]	80–95% (depending on the stage of the disease and the technique used) [17,18]	Non-invasive method. Simplicity of testing. Comprehensive tumor assessment—TNM. Disease staging [17,18].	Limited sensitivity to early stages. The risk of false-positive results. Radiation exposure. Dependence on equipment quality and interpretation [17,18].
Methyla-tyd serum septin 9	69% [19]	92% [19]	Non-invasive method. High sensitivity and specificity. Suitable for mass screening and repeat examinations. Early detection of diseases [20,21].	The sensitivity of the test in detecting precancerous conditions or early stages of cancer is lower than in advanced stages. It does not replace a full examination. A limited role in the detection of precancerous polyps. The positive mSEPT9 score was significantly higher in patients with advanced stages of CRC [21,22].

*—Diagnostic strategies recommended by the USPSTF.

**Table 2 life-15-01100-t002:** Summary table of new biomarkers used to diagnose CRC.

Novel Diagnostic Methods	Sensitivity	Specificity	Advantages	Disadvantages
Circulating tumor DNA (ctDNA) [169,170]	~70–85% (higher in advanced stages)	~90–95%	-High diagnostic accuracy: provides sensitivity and specificity in cancer detection. -Early detection: enables identification of tumors at initial stages when other methods may be less effective. -Treatment monitoring: allows assessment of therapeutic efficacy and early detection of disease recurrence. -Non-invasive procedure: blood-based analysis offers a minimally invasive alternative to tissue biopsies, enhancing patient comfort. -Molecular tumor profiling: facilitates the identification of genetic mutations, supporting personalized treatment strategies. -Assessment of minimal residual disease: useful for evaluating residual tumor burden post-treatment. -Personalized approach: development of individualized treatment strategies based on ctDNA levels.	-High cost of analysis: requires expensive equipment and reagents. -Limited sensitivity at low circulating tumor DNA (ctDNA) levels: particularly in early-stage disease or with minimal tumor burden. -Requires highly trained personnel: for accurate interpretation of results. -Potential for false-positive and false-negative results: due to technical limitations or presence of other sources of cell-free DNA. -Limited widespread availability: due to the need for specialized laboratories. -Lack of standardization: absence of universal protocols and analytical standards.
Volatile organic compounds (VOCs) [171]	~65–80%	~70–85%	-Non-invasive method: enables quick screening without invasive procedures. -Mass applicability: suitable for large-scale population screening. -Early detection: facilitates identification of cancer at initial stages. -Accessibility and simplicity of analysis: volatile organic compound (VOC) analysis can be performed in laboratory settings with relatively low costs. -Repeatability for monitoring: allows easy serial testing.	-Low specificity: false-positive results are possible due to the influence of external factors and concomitant diseases. -Limited sensitivity: especially in the early stages of the disease or low VOC levels. -Need for standardization: lack of universal protocols and standards.
Metabolomics [172]	~60–75%	~80–88%	-High sensitivity and specificity: enables the detection of disease biomarkers with high accuracy. -Early disease detection: allows the identification of pathological changes at initial stages. -Molecular characterization: facilitates understanding of characteristic metabolic alterations associated with the disease. -Potential for therapy monitoring: enable tracking of metabolic profile dynamics to assess treatment efficacy. -Non-invasive approach. -Personalized approach: supports the development of individualized treatment strategies based on metabolic profiling.	-High complexity of the analysis: requires specialized equipment and expertise. -Variability of results: depends on external factors, diet, lifestyle, and environment. -Lack of standardization: there are no universal protocols and standards. -High cost: expensive technology. -Limited sensitivity and specificity: false-positive and false-negative results are possible.
Genomics [173,174]	~70–80%	~85–90%	-High sensitivity: allows detection of diseases at the molecular level with high accuracy. -Early diagnosis: helps to identify pathologies in the early stages. -Molecular characterization: precision determination of genetic mutations. -Personalized approach: allows you to create an individual genetic profile of a tumor. -Therapy monitoring: tracking changes in genomic markers to assess the effectiveness of treatment and detect relapses. -A non-invasive or minimally invasive diagnostic method.	-High cost: requires expensive equipment and expensive analyses. -Limited standardization: lack of universal protocols and standards for all types of tests. -Long analysis cycle: the time required for conducting and interpreting genomic studies can be significant.
Exosome-based markers [175,176]	~70–85%	~85–92%	-Non-invasive diagnostic method. -High stability: exosomes protect the contents from destruction, which contributes to the high stability of biomarkers. -Information enrichment: the variety of molecules (DNA, RNA, proteins) inside exosomes allows you to obtain expanded information about the state of the source cell. -Multi-functionality: allows simultaneous analysis of different types of biomarkers for a comprehensive assessment of the body’s condition. -Personalized approach: development of an individual treatment strategy based on exosome analysis. -Early diagnosis: detection of the disease at an early stage due to the presence of specific biomarkers.	-Difficulties in standardizing and optimizing methods for isolating exosomes from biological samples. -Exosome variability, which affects the stability of the results. -High cost of analysis: expensive technologies and equipment are required for analysis -Lack of standard protocols: lack of universal methods and standards for the evaluation of exosomal biomarkers. -Limited sensitivity at low concentrations: difficulty in detecting rare or small populations of exosomes.
Proteomics [177,178]	~65–80%	~75–85%	-High information content: allows you to identify a wide range of proteins that reflect the state of the body. -Discovery of new biomarkers: promotes the discovery of previously unknown protein markers for the diagnosis of diseases. -Multifactorial analysis: provides a comprehensive assessment of pathological processes through multiple proteins simultaneously. -Early diagnosis of diseases: helps to detect changes in the protein profile in the early stages of the disease. -Personalized approach: development of customized diagnostic strategies based on proteomic profiles. -Monitoring of therapy: assessment of the effectiveness of treatment and the dynamics of the disease based on the proteomic profile.	-High complexity of the analysis: requires specialized equipment and technologies. -High cost: costs caused by the need to use expensive equipment, reagents, and the involvement of qualified specialists. -Data variability: interlaboratory differences and variations in proteomic profiles complicate standardization. -Limited sensitivity at low concentrations: difficulties in accurately identifying and quantifying proteins, especially at low concentrations.
Microbiomics [179]	~60–75%	~70–85%	-High sensitivity and specificity: allows you to detect changes in the composition of the microbiota associated with the disease. -Early diagnosis of diseases: changes in the microbiome may precede clinical manifestations. -Provides comprehensive information: reflects the state of the body and its interaction with the environment. -Non-invasive procedure: samples such as stool or saliva are analyzed, which makes the procedure less painful and more convenient. -Personalized approach: allows you to take into account the individual characteristics of the microbiome to develop personalized diagnostic and therapeutic strategies. -Monitoring the effectiveness of treatment: dynamic tracking of changes in the microbiome helps to assess the response to therapy.	-High complexity of the analysis: requires specialized equipment and technologies. -Microbiome variability: significant interindividual differences make it difficult to establish universal biomarkers. -Long study time: sequencing and analysis processes can take a long time. -High cost: significant costs for equipment, reagents, and specialists. -Standardization problems: lack of uniform standards for data collection, processing, and interpretation. -The dynamism of the microbiome: its composition can change under the influence of environmental factors, diet, and therapy, which complicates the interpretation of the results.

## Data Availability

Data are available at request from the authors.

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
