# Peer review of "Prognostic Significance of the Comprehensive Biomarker Analysis in Colorectal Cancer"

_life, 2025, doi:10.3390/life15071100_

Round 1
Reviewer 1 Report
Comments and Suggestions for Authors
The topic of the submission is interesting and it is in the focus of clinical interest. There are several good reviews in the literature on this field. This submission does not give more information now.
The ctDNA first chapter is false, the comparison of the use of serum or plasma based DNA related conclusion. It is wellknown that serum samples are not so useful in PCR based detection methods.
The methylation detection methods are already in use in the clinical practice to detect CRC, e.g. made by the South-Korean Clinomics and there are other commerically available kits. The authors did not mention theses.
There are several recent reviews on this field which are not mentioned.
There are other cell-free nucleic acids which were studied and exosomes.
Tumor cell based detection is an alternative or not?
It would be interesting to see a figure the specificity and sensitivity of the classical and modern diagnostic methods.
A table would be great to see the different methods with advantages and disadvantages.
The provided figures are too simple and primitive.
What can we expect in the future a short chapter could be beneficial.
The authors should focus on better comparison of the classical diagnostic methods and the liquid biopsy based methodology.
Author Response
Пожалуйста, прочтите приложение.

Reviewer 2 Report
Comments and Suggestions for Authors
This article reviews the prognostic significance of comprehensive biomarker analysis in colorectal cancer. Extracellular DNA (ecDNA) present in the blood of cancer patients has been utilized as a diagnostic biomarker since the 21st century. Circulating tumor DNA (ctDNA), a component of extrachromosomal DNA (ecDNA) containing tumor-specific genetic mutations, shows promise for early cancer detection and treatment monitoring.
ctDNA serves as a crucial biomarker for predicting postoperative recurrence risk, particularly in high-risk stage II/III colorectal cancer (CRC) patients. Volatile organic compounds (VOCs) have emerged as novel biomarkers, with breath analysis showing potential for cancer diagnosis and monitoring of recurrence through techniques such as gas chromatography-mass spectrometry (GC-MS) and electronic nose (E-nose)3. Metabolomics studies have identified 635 metabolites associated with CRC, notably L-phenylalanine and linoleic acid.
Genetic and epigenetic analyses revealed 1,910 distinct mutations in 151 patients, including pathogenic variants in 9 genes (e.g., APC, ATM, BRCA1). Epigenetic modifications, such as DNA methylation, also influence gene expression. Molecular classification identified the CMS4 subtype as responding well to irinotecan-based chemotherapy, while CMS1 showed the poorest prognosis and CMS2 the best outcomes.
Novel biomarkers include exosomes involved in cell communication and stool-based immune-related proteins. While this review provides valuable insights, two concerns emerge:
- Cost-effectiveness analysis is lacking for new biomarkers (e.g., potential cost benefits of VOC/immune protein-based early diagnosis vs advanced-stage treatment)
- AI applications are not addressed, particularly multimodal diagnostics combining ctDNA, immune proteins, and VOCs to enhance accuracy.
I would appreciate your consideration of these points.
Author Response
Пожалуйста, посмотрите приложение.

Round 2
Reviewer 1 Report
Comments and Suggestions for Authors
The quality of the submission was improved.
The iThenticate score is 36%, so you have to reduce it under 30%.
We do not agree on the use of serum or plasma, you cited Heintzel's work, but it is not acceptable. Please use more accepted article (from internationally accepted researcher) or a society recommendation as a source. This fact is wellknown in the community where they work with cell-free nucleic acids.
